# The Role of Sentinel Lymph Node Biopsy in Breast Cancer Patients Who Become Clinically Node-Negative Following Neo-Adjuvant Chemotherapy: A Literature Review

**Giulia Ferrarazzo** [1,*], **Alberto Nieri** [2], **Emma Firpo** [3], **Andrea Rattaro** [3], **Alessandro Mignone** [1], **Flavio Guasone** [3], **Augusto Manzara** [1], **Giuseppe Perniciaro** [4] **and Stefano Spinaci** [5]

1   Nuclear Medicine, Ospedale Villa Scassi ASL3, 16149 Genova, Italy;
    alessandro.mignone@asl3.liguria.it (A.M.); augusto.manzara@asl3.liguria.it (A.M.)
2   Nuclear Medicine Unit, Oncological Medical and Specialist Department, University Hospital of Ferrara,
    44124 Cona, Italy; a.nieri@ospfe.it
3   Breast Surgery, Department of Surgery, Ospedale Villa Scassi ASL3, 16149 Genova, Italy;
    emma.firpo@asl3.liguria.it (E.F.); andrea.rattaro@asl3.liguria.it (A.R.); flavio.guasone@asl3.liguria.it (F.G.)
4   Division of Plastic and Reconstructive Surgery, Burn Unit, Ospedale Villa Scassi ASL3, 16149 Genova, Italy;
    giuseppe.perniciaro@asl3.liguria.it
5   Breast Unit, Department of Surgery, Ospedale Villa Scassi ASL3, 16149 Genova, Italy;
    stefano.spinaci@asl3.liguria.it
*   Correspondence: giulia.ferrarazzo@asl3.liguria.it; Tel.: +39-010-8492299

**Abstract:** Background: In clinically node-positive (cN+) breast cancer (BC) patients who become clinically node-negative (cN0) following neoadjuvant chemotherapy (NACT), sentinel lymph node biopsy (SLNB) after lymphatic mapping with lymphoscintigraphy is not widely accepted; therefore, it has become a topic of international debate. Objective: Our literature review aims to evaluate the current use of this surgical practice in a clinical setting and focuses on several studies published in the last six years which have contributed to the assessment of the feasibility and accuracy of this practice, highlighting its importance and oncological safety. We have considered the advantages and disadvantages of this technique compared to other suggested methods and strategies. We also evaluated the role of local irradiation therapy after SLNB and state-of-the-art SLN mapping in patients subjected to NACT. Methods: A comprehensive search of PubMed and Cochrane was conducted. All studies published in English from 2018 to August 2023 were evaluated. Results: Breast units are moving towards a de-escalation of axillary surgery, even in the NACT setting. The effects of these procedures on local irradiation are not very clear. Several studies have evaluated the oncological outcome of SLNB procedures. However, none of the alternative techniques proposed to lower the false negative rate (FNR) of SLNB are significant in terms of prognosis. Conclusions: Based on these results, we can state that lymphatic mapping with SLNB in cN+ BC patients who become clinically node-negative (ycN0) following NACT is a safe procedure, with a good prognosis and low axillary failure rates.

**Keywords:** sentinel lymph node biopsy; breast cancer; neoadjuvant chemotherapy; surgery



## 1. Introduction

Sentinel lymph node (i.e., the first lymph nodes in which cancer cells appear, which are most likely to spread from the primary neoplasm) biopsy (SLNB) is a minimally invasive procedure that has largely replaced axillary lymph node dissection (ALND) in early breast cancer (BC) surgery, especially in the case of no axillary lymph node involvement (cN0) and in patients at stage T1-2 cN0 early BC with one to two positive sentinel lymph nodes (SLNs). The advantages of this technique are its minimally invasive approach to the axilla and the possibility of avoiding the complications of ALND [1–5].

Neoadjuvant chemotherapy (NACT) plays a proven role in BC management [6]. Once reserved for unresectable locally advanced disease patients, today it is the gold standard of treatment in 10% of BC patients, with a cytoreductive and curative aim. In advanced locoregional BC, NACT represents an opportunity to reduce the extension of primary neoplasia and to down-stage in the case of axillary lymph node involvement (cN+), leading to more axillary-conservative surgery in comparison to ALND and its possible complications (shoulder stiffness, arm lymphedema, numbness, paresthesia, chronic pain, limited movement, lymphangitis, and tissue fibrosis) [7–15].

NACT candidate patients are a heterogenous group, and many schemes of NACT are proposed based on the molecular characteristics of the neoplasm [16]. For Luminal A BC (Estrogen Receptor (ER)-positive and Progesterone Receptor (PgR)-positive) cases, usually only endocrine therapy (ET) is recommended. For Luminal B BC cases (ER-positive and/or PgR-positive and Human Epidermal Growth Factor Receptor Type 2 (HER-2)-negative), standard therapy is an association between ET and chemotherapy (ChT). For Luminal B–HER-2-positive cases, ET plus ChT plus anti-HER-2 therapies (trastuzumab $\pm$ pertuzumab) are recommended. Finally, for triple-negative breast cancer (TNBC), that presents no expression of both ERs, PgRs, and the HER-2 protein, usually only ChT is recommended [17,18].

Patients can globally achieve a pathological complete response (pCR) in 20% to 80% of cases depending on the characteristics of primary malignancy [16,19,20]. The literature reports a complete axillary lymph node response rate of >50% in TNBC and 80% in HER-2-positive BC [21–23]. The relationship between the pCR and increase in overall survival (OS) finds less evidence in high-proliferative-index Luminal B and progesterone (PgR)-negative cases. A lesser correlation was shown for Luminal A [21–23].

Unfortunately, in many cases, NACT can damage lymphatic vessels due to inflammation or fibrosis induced by the treatment, resulting in an anatomical modification of the lymphatic drainage system. In addition, a heterogeneous tumor regression with a different degree of response in the axillary LNs can occur [24–27]. These events may result in a difficult and incorrect SLN mapping, increasing the false negative rate (FNR) for SLNBs [25,26,28].

The current guidelines do not agree on the best strategy. German, Austrian, and Scandinavian guidelines recommend ALND after NACT in cN+ patients. On the contrary, the European Society of Medical Oncology (ESMO) and the National Comprehensive Cancer Network (NCCN) in the USA recommend SLNB and the excision of a minimum of three sentinel nodes [17,29]. In countries such as Italy, Denmark, Russia, and Hungary, SLNB or targeted axillary dissection (TAD) are endorsed as the first choice to stage the axilla in this group of patients, while the German Breast Committee of the Working Group for Gynecological Oncology (AGO Breast Committee) TAD and ALND are considered equivalent methods in cases with ≤3 suspicious nodes at diagnosis and a good NACT response (ycN0). In patients with ≥4 suspicious nodes, however, ALND is the preferred technique [30,31].

As a result, the indications of the SLNB and its oncological consequences in patients treated with NACT (both cN0 and cN+ at staging) are debated.

For many years, the SLNB has been performed in patients with no evidence of axillary involvement at staging who were submitted to surgery after NACT with FNRs of about 10% and acceptable SLN identification rates [21,32–35]. Conversely, until the last decade, patients with verified axillary involvement at staging were submitted to ALND, even in cases of complete response at this level [36,37].

Recently, prospective studies on large samples of patients have begun to change this concept, indicating an overall SLN identification rate (SNIR) ranging from 87.6 to 97.2%, but FNRs higher than the accepted 10% are indicated by previous studies [37–40].

In order to minimize the SLNB FNR in patients with axillary involvement, different techniques and strategies have been proposed (e.g., TAD by marking biopsy-positive lymph nodes prior to NACT with various tools such as carbon suspension, metallic clip, radioactive or magnetic seeds, radar reflectors, and radiofrequency markers, the use of a

dual tracer, or the removal of at least three lymph nodes) [41–45]. To date, the prognostic significance of these strategies has not been sufficiently investigated, but some studies have demonstrated, and confirmed with follow-up data, that the SLNB is neither related with an increased recurrence rate in axilla, nor with a worsened disease-free survival (DFS) or OS [5,46–50].

Ongoing studies aim to investigate the preferable procedure for axillary surgery following NACT in the case of initially confirmed axillary lymph node metastases. For example, the AXSANA (AXillary Surgery After NeoAdjuvant Treatment) study (NCT04373655), conducted by the European Breast Cancer Research Association of Surgical Trialists (EU-BREAST), is a multinational prospective cohort study designed to better clarify this topic [51].

This trial, which enrolls cN+ patients with no evidence of axillary metastases after NACT, intends to assess the impact of different surgical strategies on the oncologic outcome, life quality, and arm morbidity [50,52].

On this basis, this literature review aims to analyze the current state of the art of SLNB alone in initially cN+ BC patients who become cN0 following NACT, focusing on the many studies which have been conducted in the last six years, contributing towards highlighting the importance and oncologic safety of this practice.

## 2. Methods

A systematic literature review was conducted on PubMed, Scopus, and Cochrane databases between January 2018 and August 2023. As previously anticipated, the aim of this research was to analyze papers, published in the last six years, describing the role of SLNB in cN + BC patients who result in cN0 after neoadjuvant treatment regarding surgical management.

The search string (either as text or MeSH) was "breast" AND "sentinel" AND "neoadjuvant". Overall, we identified 640 articles from 2018 to 2023 (the last search was performed on 1 August 2023). Two authors performed an independent review of the related abstracts (G.F. and A.N.), while a third author was consulted in case of discrepancies (S.S.). No language restriction was applied to the search, but only articles in English were evaluated. In addition, the more pertinent studies in the articles' references were reviewed. In accordance with the Preferred Reporting Items for Systematic Review and Metanalysis (PRISMA) criteria, after the exclusion of duplicates and articles not meeting the inclusion criteria or not in the area of interest, 20 studies were finally included in the systematic review. The selected papers were examined, summarized, and described in relation to their adherence to the topic. The inclusion criteria were original articles, clinical studies, and clinical trials (randomized, prospective, or retrospective). The exclusion criteria were editorials, letters, case reports, and series including fewer than 5 patients. Figure 1 provides a graphic illustration of the search and review strategy.

Through the same databases, with the purpose of widening this topic, we performed a further literature search regarding the role of local irradiation therapy after SLNB, the state of the art of SLN mapping techniques in these kinds of patients, and, finally, the alternative techniques proposed to minimize SLNB FNR. Because of the wide range of existing literature focused on these topics, we did not perform a systematic review for these sections but rather a narrative description.

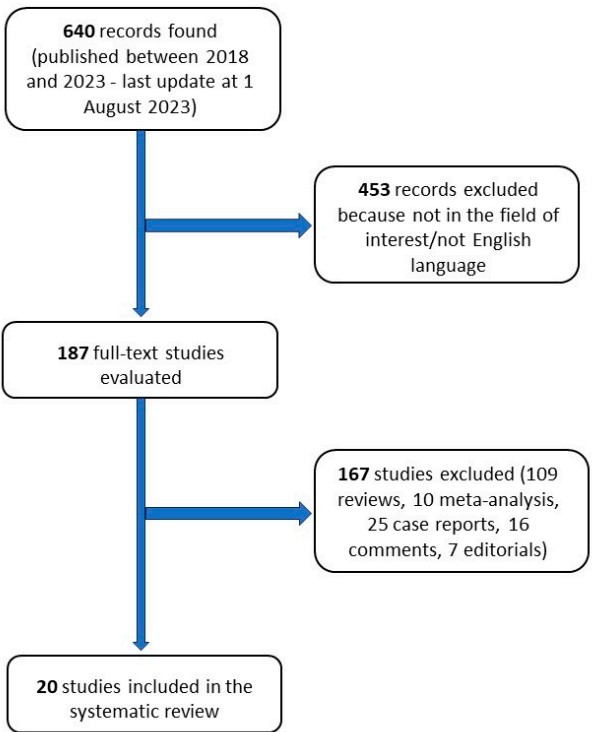

**Figure 1.** Flow of records searched according to the present systematic review.

## 3. SLNB Role in the Surgical Management of cN+ Patients Resulted cN0 after NACT

Over the past six years, several studies have been published to evaluate the effectiveness and outcomes of SLNB in the surgical management of cN+ patients who resulted in being cN0 after NACT. Different aspects of these studies were analyzed.

### 3.1. Feasibility, SLNs Identification Rate, and False Negative Rate

Ogawa et al.'s study consisted of 105 women; after chemotherapy, 80 of these patients became or remained cN0. Axillary management was determined using SLNB in 53 of these patients (20 patients with positive SLNs were submitted to ALND), while the remaining patients were directly submitted to ALND. Recurrence-free survival (RFS) and OS rates demonstrated no statistically significant difference between the two groups. Researchers concluded that SLNB after NACT could be a safe procedure even in this group of patients [53].

Classe et al. evaluated FNR, accuracy, and safety with the study "Ganea 2" and estimated an overall SLNB FNR of 11.9% in patients down-staged as cN0 from cN1 (increased up to 19.3% when only one SLN was removed) [54].

In their study, Choi et al. identified patients with negative SLN not submitted to ALND (group A), patients with negative SLN submitted to ALND (group B), and patients submitted to ALND with no evidence of metastasis on pathology (group C). They calculated SNIR and FNR with a median number of five removed SLNs. These three groups showed no significant differences in axillary RFS, DFS, or OS [55].

Berberoglu et al. investigated the diagnostic value of SLNB, and an SLN was identified in 92.6% of subjects. The authors assessed the sensitivity, specificity, and accuracy of the technique to predict macro-metastasis (85.7, 86.5, and 86.2%, respectively), omitting FNR. Researchers concluded that SLNB using the radioisotope (RI) method for mapping seems to be efficient in differentiating patients who need ALND, even after NACT [56].

Recently, Jimenez-Gomez et al. detected a percentage of SLNB FNR < 10%, considered to be the minimum admitted in the literature [57].

### 3.2. Effect of SLNB Procedures on Local Recurrence

In a study conducted by Piltin et al., 159 initially cN1/2 patients underwent SLNB alone, and only 1 patient developed axillary lymph node recurrence. The authors evidenced no statistical difference in the 2-year freedom-from-regional-recurrence rate between patients submitted to SLNB alone vs. ALND (99.1% and 96.4%, *p* = 0.10, respectively). Therefore, patients with complete response after NACT who underwent SLNB alone did not show worse oncologic outcomes compared to those who underwent ALND [49].

Similarly, in an analysis conducted on 58 cN1/2 patients submitted to NACT, that resulted in a complete clinical response and who performed SLNB alone, Wong et al. reported no axillary lymph node recurrence during a follow-up of five years [58].

Other studies published in 2021 showed that a conservative approach was feasible and apparently did not change disease control and oncological outcomes since they reported very low or no recurrences in the axilla [59–61].

In the same way, Barrio et al. examined a cohort of 769 cN+ patients treated with NACT, and they highlighted that patients with axillary complete response who performed SLNB alone (234 patients with ≥3 negative SLNs) had low nodal recurrence rates. These results support omitting ALND in such patients [47].

Recently, Tinterri et al. analyzed both cN0 patients (131) and cN+ patients (160) who achieved a complete nodal response after NACT and were subjected to SLNB. In this setting, SLNB proved to be an acceptable procedure because of a good prognosis and low axillary failure rates in both groups of patients. No significant differences in oncological outcomes were found in the case of both axillary involvement or non-involvement after NACT, neither in the SLNB alone nor in the SLNB + ALND group [5].

### 3.3. Effect on Outcome

In their multicentric study, Cabioglu et al. analyzed 303 cytopathology-proven cN+ patients, 211 of whom reached a complete response after NACT. The authors observed that some factors such as breast and/or nodal complete response, cT1-2 stage, or low-volume residual nodal disease with luminal pathology affected local outcome. They proved that ALND could be avoided in patients treated with NACT who underwent SLNB, as long as axillary radiotherapy is provided [62].

In 2023, Galimberti et al. retrospectively recruited 466 cN0 patients and 222 cN1/2 patients. After a complete nodal response after NACT, they were submitted to surgery (SLNB alone or plus ALND). The 10-year cumulative incidence of distant events for initial cN1/2 patients and initial cN0 patients (16.6% vs. 13.1%) did not show statistical significant differences (*p* = 0.148) [50,63].

Tercan et al. evaluated the oncologic safety of SLNB in BC patients who were then submitted to NACT, with the pathologically and/or clinically and radiologically metastatic involvement of axillary lymph nodes at the time of diagnosis [64].

As a result of this study, Martelli et al. wished to evaluate the outcomes in this setting of patients. They assessed OS and DFS in the case of SLNB alone in cN0 patients (81) versus SLNB + ALND in cN1 patients (272) and, during a 10-year follow-up, OS and DFS of the SLNB-alone group presented no statistical differences compared to the SLNB + ALND group, with no axillary recurrences in the SLNB-alone group [48].

A single-center retrospective study by Kim et al. compared OS, DFS, and incidence of postoperative complications, such as lymphedema and shoulder stiffness, in the case of SLNB alone or plus ALND. The five-year OS and DFS rates showed no significant differences in the two groups. Instead, a significantly higher number of ALND experienced post-operative complications with respect to patients who underwent SLNB alone, especially lymphedema. The researchers concluded that SLNB is a suitable technique to use in patients who reach a complete response after NACT [65].

Kahler-Ribeiro-Fontana et al. confirmed that SLNB is a safe procedure to use in this setting of patients as it does not have any negative impact on the outcomes. In this study, 5- and 10-year OS and DFS in cN+ patients before NACT showed no statistical differences

compared to cN0 patients before NACT. Furthermore, patients who avoided ALND had a better outcome with respect with those who underwent ALND [46].

### 3.4. Requirements Regarding the Number of SLNs Retrieved

Sharp et al. showed that, despite using a single tracer without TAD, locoregional recurrence events remained low, and DFS did not result in being significantly different by removing 1 vs. 2 vs. ≥3 SLNs which proved negative at histology [66].

Galimberti et al. also found that there were no requirements for the number of SNs retrieved [50].

### 3.5. Effects on Clinical Practice

Nguyen et al. noted an increase in the use of SLNB (±ALND) in a population studied between 2009 and 2017 (from 28% at the beginning of the study up to 86% at the last observation), with a concomitant decrease in ALND procedures (from 100% to 38%). ALND was avoided in 48% of patients submitted to SLNB, and no evidence of nodal recurrence was described at short-term follow-up [67].

### 3.6. Effect on Complications

Choi et al. observed lymphedema and arm movement morbidity in 7.1% of patients undergoing SLNB alone and 27.3% in patients treated with ALND [55].

Kim et al. found a significantly higher complications rate, in terms of shoulder stiffness and arm lymphedema, in patients submitted to ALND compared to those who were treated with SLNB alone (16.0% vs. 34.1%) [65].

In general, these studies suggest that SLNB is a suitable procedure even in patients who received NACT prior to surgery and that it can contribute to the decreased use of more invasive procedures and their complications, with no significant differences in the outcomes.

Possible limitations of this review are the inclusion of trials with mixed BC patient populations and the analysis of papers published in the last six years.

In Tables 1–3, the most important characteristics and results extracted from all the studies included in the present review are presented.

**Table 1.** General information of the included studies regarding surgical management.

| Author | Year | Country | Study Design | Number of Involved Centers | Funding Sources |
|---|---|---|---|---|---|
| Choi et al. [55] | 2018 | South Korea | P | Single | No |
| Nguyen et al. [67] | 2018 | USA | R | Single | No |
| Ogawa et al. [53] | 2018 | Japan | R | Single | No |
| Classe et al. [54] | 2019 | France | P | Multicentric | Yes |
| Berberoglu et al. [56] | 2020 | USA | R | Single | None declared |
| Piltin et al. [49] | 2020 | USA | R | Single | None declared |
| Barrio et al. [47] | 2021 | USA | R | Single | Yes |
| Cabioğlu et al. [60] | 2021 | Turkey | R | Multicentric | Yes |
| Damin et al. [60] | 2021 | Brazil | R | Single | No |
| Kahler-Ribeiro-Fontana et al. [41] | 2021 | Italy | R | Single | Yes |
| Kim et al. [65] | 2021 | South Korea | R | Single | No |
| Lee et al. [61] | 2021 | South Korea | R | Single | Yes |
| Riogi et al. [59] | 2021 | UK | P | Single | No |

**Table 1.** *Cont.*

| Author | Year | Country | Study Design | Number of Involved Centers | Funding Sources |
|---|---|---|---|---|---|
| Sharp et al. [66] | 2021 | USA | R | Single | Yes |
| Wong et al. [58] | 2021 | Canada | R | Single | None declared |
| Martelli et al. [48] | 2022 | Italy | P | Single | None declared |
| Tercan et al. [64] | 2022 | Turkey | R | Single | None declared |
| Galimberti et al. [50] | 2023 | Italy | R | Single | No |
| Jimenez-Gomez et al. [59] | 2023 | Spain | R | Single | No |
| Tinterri et al. [5] | 2023 | Italy | R | Single | No |

P: prospective; R: retrospective.

**Table 2.** Patient key characteristics, clinical setting, and index test key characteristics of studies regarding surgical management.

| Author | Enrollment | Patient Stage | N. of cN+ Patients → cN0 after NACT | N. of SLNB Alone | Mean Age (Years) | Axillary Staging | SLN Mapping Technique | Median/Mean of Retrieved Nodes | Median FU (Months) |
|---|---|---|---|---|---|---|---|---|---|
| Choi et al. [55] | 2007–2014 | cT1-T4, N1-3 | 506 | 85 | 44.4 ± 9.3 | SLNB | RI and BD | 5 (2–9) | 51 |
| Nguyen et al. [67] | 2009–2017 | cT0-T4, N1 | 430 | 93 | 50.5 | SLNB | RI and BD | 2 (1–9) | 9 |
| Ogawa et al. [53] | 2006–2015 | cT1-T4, N0-3 | 48 | 33 | 52.6 | SLNB | BD | 2.4 | 59 |
| Classe et al. [54] | 2010–2014 | cT1-4, N0-N2 | 351 | 1 | 52 | SLNB | RI and BD | 2 (1–8) | 36 |
| Berberoglu et al. [56] | / | cT0-4, N0-N2 | 91 | 87 lesions | 47 | SLNB | RI | 1.0–4.0 | |
| Piltin et al. [49] | 2009–2019 | cT1-4, N1-3 | 602 | 159 | 45 | SLNB | Not specified | 3 (1–12) | 34 |
| Barrio et al. [47] | 2013–2019 | cT1-3, N1 | 555 | 234 | 49 | SLNB | RI and BD | 4 (3–5) | 40 |
| Cabioğlu et al. [60] | 2004–2018 | cT1-4, N1-N3 | 303 | | 46 | SLNB | RI and BD | 3 | 36 |
| Damin et al. [60] | 2010–2016 | cT1-4, N1-N2 | 59 | 38 | 49.08 ± 0.84 | SLNB | RI and BD | 2 | 55.8 |
| Kahler-Ribeiro-Fontana et al. [41] | 2000–2015 | cT1-3, N0-N2 | 222 | 131 | 45 | SLNB | RI | 2 (1–6) | 110 |
| Kim et al. [65] | 2006–2015 | cT1-4, N1-3 | 223 | 94 | 46 | SLNB | RI and BD | 2.2 ± 1.2 | 57 |
| Lee et al. [61] | 2003–2014 | cT1-T4, N1-3 | 242 | 760 | 45.1 | SLNB | RI | 4.9 ± 2.6 | 60 |
| Riogi et al. [59] | 2007–2016 | cN0-N+ | 56 | 40 | 50 | SLNB | RI and BD | 2 (1–7) | |
| Sharp et al. [66] | 2004–2018 | cT1-3, N0-N2 | 68 | 68 | 50 | SLNB | RI and BD | 1- ≥3 | 46.8 |
| Wong et al. [58] | 2013–2018 | cT1-3, N0-N2 | 132 | 102 | 50 | SLNB | RI and BD | 3 (2–4) | 36 |
| Martelli et al. [48] | 2007–2015 | cT2, N0-N1 | 121 | 81 | 47 | SLNB | RI | 2 (1–8) | 108 |
| Tercan et al. [64] | 2013–2020 | cT1-4, N1-2 | 90 | 44 (39 ypCR + 6 ypNCR) | 49.6 | SLNB | RI and BD | ≥3 | 34 ± 18 |
| Galimberti et al. [50] | / | cT1-3, N0-N2 | 222 | 222 | 45 | SLNB | RI | 2 (1–6) | 120 |
| Jimenez-Gomez et al. [59] | 2010–2017 | cT1b-T4, N+ | 168 | 48 | | SLNB | RI and BD | ≥2 | 60 |
| Tinterri et al. [5] | 2008–2021 | cT1-4, N0-N+ | 160 | 100 | 50 | SLNB | RI | 1 | 50 |

BD: blue dye; FU: follow-up; NACT: neo-adjuvant chemotherapy; RI: radioisotope; SLN: sentinel lymph node; SLNB: sentinel lymph node biopsy; ypCR: pathologic complete response; ypNCR: pathologic near-complete response.

**Table 3.** Aims and outcomes of the included studies regarding surgical management.

| Author | Aim of the Study | SNIR (%) | FNR (%) | OS (%) | DFS (%) | Axillary Recurrence (%) | Outcome |
|---|---|---|---|---|---|---|---|
| Choi et al. [55] | Evaluate feasibility of SLNB | 98.3 | 7.5 | 92.9 | 81.2 | 2.0 | SLNB can be feasible and oncologically safe |
| Nguyen et al. [67] | Evaluate effect of SLNB in clinical practice | / | 5 | / | / | 0.0 | Significantly increased use of SLNB alone |
| Ogawa et al. [53] | Assess effectiveness, SNIR, and FNR of SLNB | 94.3 | / | 80.0 | 60.0 | 30.0 | SLNB does not affect the axillary failure rate or the prognosis |
| Classe et al. [54] | Assess diagnostic accuracy and safety of SLNB | / | 11.9 | / | / | / | For SLNB alone, an accurate selection of post-NACT negative SLN patients is necessary |
| Berberoglu et al. [56] | Evaluate diagnostic value of SLNB | 92.6 | 5.7 | / | / | / | SLNB is feasible and efficient |
| Piltin et al. [49] | Compare SLNB alone vs. ALND | / | 3.8 | / | 97.4 | 0.9 | SLNB alone is not oncologically inferior to ALND during a short-term FU period |
| Barrio et al. [47] | Assess axillary LN recurrence | / | / | / | 92.7 | 1 | If ≥3 negative SLNs with SLNB alone, axillary LRR is low |
| Cabioğlu et al. [60] | Evaluate factors affecting local recurrence and overall outcome | / | / | / | 88.0 | 1.1 | ALND could be avoided in selected patients |
| Damin et al. [60] | Evaluate safety of SLNB, efficacy, and oncological outcomes | 93.2 | <10 | 89.0 | 82.0 | 2.6 | SLNB could be successfully used and does not compromise disease control and oncological outcomes |
| Kahler-Ribeiro-Fontana et al. [46] | Assess axillary LN recurrence, OS, DFS | / | / | 84.8 | 81.4 | 1.6 | SLNB alone is acceptable and not associated with a worse outcome |
| Kim et al. [65] | Evaluate safety, axillary LN recurrence rate, and incidence of side effects | / | 10 | 96.3 | 94.2 | 1.1 | SLNB is oncologically safe |
| Lee et al. [61] | Evaluate prognosis and oncological outcomes of SLNB alone | / | / | 93.0 | 98.0 | 2.0 | SLNB alone is associated with low LRR |
| Riogi et al. [59] | Evaluate management of the axilla and outcomes | / | / | 79.4 (of 165 pts) | 24.0 (of 165 pts) | 0.0 | Acceptable outcomes of conservative approach in the axilla after NACT |
| Sharp et al. [66] | Assess LRR rate for SLNB | / | / | 85.0 | 85.0 | 3.0 | Low LRR events and DFS statistically similar between SLNs number |
| Wong et al. [58] | Evaluate oncological safety of SLNB | 96.9 | / | / | / | 5.9 | SLNB alone is associated with low and acceptable short-term axillary recurrence rates |
| Martelli et al. [48] | Assess feasibility of SLNB, OS, DFS | / | / | 89.0 | 79.0 | 0.0 | SLNB is oncologically safe |
| Tercan et al. [64] | Evaluate efficiency and safety of SLNB | / | / | 92.3 in ypCR and 100 in ypNCR | / | 0.0 | No event developed in cases with ypCR and ypNCR in the breast and axilla |
| Galimberti et al. [50] | Assess axillary LN recurrence, incidence of distant events | / | / | / | / | 1.8 | SLNB alone demonstrate no worse outcome in cN+ patients who became cN0 after NACT |
| Jimenez-Gomez et al. [59] | Evaluate feasibility and diagnostic accuracy of SLNB | / | 7 | / | 41.4 | <1 | SLNB provides useful and reliable information about cancer staging, leading to a decrease in possible arm morbidity |
| Tinterri et al. [5] | Compare the characteristics and oncological outcomes of SLNB in cN0 and cN+ patients before NACT and axillary surgery | / | / | 93.2 | 83.6 | 1.3 | SLNB shows good prognosis and low axillary failure rates in cN0 or cN+ patients undergoing NACT who subsequently remained or became cN0 |

ALND: axillary lymph node dissection; DFS: disease-free survival; FNR: false negative rate; FU: follow-up; LN: lymph node; LRR: locoregional recurrence; NACT: neo-adjuvant chemotherapy; OS: overall survival; SLN: sentinel lymph nodes; SLNB: sentinel lymph node biopsy; SNIR: sentinel lymph nodes identification rate; ypCR: pathologic complete response; ypNCR: pathologic near-complete response.

## 4. Role of Local Irradiation Therapy after SLNB in cN+ Patients Who Resulted cN0 after NACT

Another topic of discussion is adjuvant locoregional radiotherapy (RT) following SLNB either after mastectomy or conserving surgery, which shows a lack of standardization. Moreover, the additional axillary management (ALND vs. RT) of patients with a positive SLNB after NACT is a decision made by a multidisciplinary team [50,57,68].

Many studies have proven that axillary RT after SLNB is a safe alternative to ALND in patients submitted to primary surgery with 1–2 positive nodes.

The Intergroup Study EORTC 10981-22023 (After Mapping of the Axilla: Radiotherapy Or Surgery?—AMAROS) compared ALND and lymph node irradiation in patients with a positive SLNB. It demonstrated that the control of local disease is equivalent and highlighted low morbidity in the RT group [69]. In the same way, a meta-analysis conducted by the Early Breast Cancer Trialists' Collaborative Group (EBCTCG) evidenced a lowered risk of loco-regional recurrence (LRR) at 10 years in the case of nodal radiotherapy in the presence of 1–3 positive nodes [70].

Other authors have attempted to assess the role of local irradiation in patients with complete response or who resulted in being ycN0 after NACT. Miyashita et al. [71] highlighted no differences in locoregional recurrence free survival (LRRFS) and OS between patients who resulted in being ycN0 after NACT and who were submitted or not to local irradiation. Likewise, Van Hemert et al. stated that RT reduces LLR in patients with complete response after NACT, but this could cause early and late toxicity (regional pain, fibrosis, swelling, and breast morphological alteration are described in up to 40% of cases), with a consequent reduction in quality of life [72,73], and it does not improve OS [74].

In this scenario, some studies have attempted to assess the role of adjuvant RT in patients with SLNB who resulted in being negative. Cho et al. [75] described no differences in 5-year DFS and OS even in this setting of patients who were subsequently treated or not by RT. Kim et al. [76] evidenced no statistically significant differences in 5-year LRRFS and described that RT improves LRRFS only in TNBC patients; for these reasons, the optimal combination of surgery and RT in this context is still up for debate [68]. There is a great expectation for prospective data to be presented by ongoing studies, for example the NSABPB-51/RTOG1304 trial [77] and the Alliance clinical trial [78], from the perspective of a possible de-escalation of radiotherapy for such patients.

In conclusion, based on the evidence that RT might not improve OS in patients who resulted in being cN0 after NACT, the potential relevance of SLNB on the performance of adjuvant RT is less clear; therefore, further studies are underway to verify a possible reduction in the use of this procedure as well.

## 5. State of the Art of SLN Mapping Techniques in Patients Subjected to NACT

Many studies encourage SLNB mapping, even in patients who are submitted to surgery after NACT.

SLN mapping is usually performed using radioisotope (RI) or blue dye (BD) methods alone or in combination. However, some issues related to this context (i.e., high costs, few centers with nuclear medicine facilities, potential allergic reactions to the tracer, etc.) have required new tracers to be studied, such as indocyanine green (ICG), superparamagnetic iron oxide (SPIO), and carbon nanoparticles (CNPs) [79–82].

Most of these alternative approaches share a similar technique: tracers are injected subcutaneously or subdermally in the breast, reaching the lymph nodes via the lymphatic system. Subsequently, SLNs marked with BD or CNPs are visually identified in the surgical field, SLNs marked with RI are revealed via a gamma probe (GP), SLNs marked with SPIO, which is blocked in sinuses and macrophages, are detected using a magnetometer (MM), and SLNs marked with ICG are revealed via a fluorescence imaging system (FIS).

Some studies have attempted to evaluate the best technique to use in those patients who underwent SLNB after NACT (Table 4).

**Table 4.** Studies about SLNB mapping and technique of choice.

| Author | Year | Patients | SLNB Mapping Technique | Detection Rate | Comments |
|---|---|---|---|---|---|
| Chirappapha et al. [83] | 2020 | 21 | RI<br>BD<br>ICG | 53.87%<br>81.78%<br>93.22% | Every combination demonstrated a good performance. |
| Giménez-Climent et al. [84] | 2021 | 89 | RI<br>SPIO | 97.8%<br>97.8% | This study demonstrated a non-inferiority of SPIO compared to RI. |
| Sun et al. [85] | 2023 | 123 (2 patients after NACT) | CNPs<br>CNPs plus BD<br>BD plus ICG | 97.4%<br>97.6%<br>95.5% | This study proved that, despite the lack of patients treated with NACT, these techniques could be valid even in this setting of patients. |

BD: blue dye; CNPs: carbon nanoparticles; ICG: indocyanine green; NACT: neoadjuvant chemotherapy; RI: radioisotope; SPIO: superparamagnetic iron oxide.

In conclusion, these studies globally demonstrated that the use of both gold standard tracers and techniques (RI and BD) and other tracers (ICG, SPIO, or CNP), alone or in combination, may be appropriate and could represent a good alternative for centers where performing SLNB according to the gold standard is not possible.

## 6. Alternative Techniques and Strategies Proposed to Minimize SLNB FNR

A critical issue of SLNB alone, in the case of a negative result at histological analysis, is related to a lack of information regarding residual disease, which is fundamental to program adjuvant therapy. Hypothetically, these data could be derived from the possible non-response of the primary tumor [50], but the minimization of SLNB FNR in initially cN+ patients appears to be a major problem.

Different techniques and strategies have been proposed over the years. Among them, the most prominent is targeted axillary dissection (TAD), whose main goal is lower FNR with respect to SLNB. TAD consists of the combination of SLNB and targeted lymph node biopsy (TLNB), that is, the removal of the most suspicious malignant lymph node which resulted in being metastatic prior to NACT. Numerous and different marking systems have been developed to label lymph nodes for eventual TAD: positive axillary nodes could be marked by metallic clips, usually in titanium, radioactive, or magnetic seeds, and radar reflectors and radiofrequency markers or carbon particles through their ultrasound (US)-guided insertion in the selected lymph node before chemotherapy [41,44,86]. During surgery, target lymph nodes can be revealed by different systems, such as imaging (usually intra- or preoperative US) or specific probes based on the chosen marker [52].

However, such methods have some disadvantages related to national regulations prohibiting the use of radioactive markers in some healthcare facilities, the difficulty of establishing the exact number of lymph nodes to be marked, and the possibility of not finding an eventual clip in the operating field (up to 30% of cases). Further issues are high costs and a great expenditure of time.

Going into more detail, the use of clips/coils in TAD is the most tested technique [31,42,87,88]. Some of its advantages are no use of radioactivity and no relevant artifacts at magnetic resonance. An important limit is that clips/coils cannot be detected by means of probes but exclusively by imaging techniques, mostly US. Therefore, the detection rate depends on the ability to locate the clip/coil. Following its identification, a further marking procedure is usually carried out using a wire or a marker detectable through a probe. However, if the clip/coil is easily visible, its identification can proceed directly using intraoperative US [31]. Unfortunately, the detection rate is relatively low in these cases (about 70% in the largest available dataset), and some instruments have some disadvantages; for example, some clips demonstrate a decrease in visibility over time (e.g., hydrogel clips) and could cause an inflammatory reaction of the node tissue (especially

in case of hydrogel-containing clips), which may be misinterpreted on pathological examination, and even a possible, although rare, allergic reaction (especially in the case of nickel-containing clips). Another problem is the approval of some clip systems explicitly for breast lesions but not for lymph nodes [52].

Another marker for TAD/TLNB consists of the US-guided injection of highly purified carbon particles (0.1–0.5 mL solution) into the selected lymph node and the surrounding soft tissue [89]. Some disadvantages are the impossibility of visualizing such a marker without a surgical exploration (no imaging technique can detect carbon particles), and the migration of carbon particles and skin discoloration are possible [90,91].

Among markers which take advantage of probe-guided detection, there is the US-guided insertion of titanium-encapsulated radioactive seeds into the suspected lymph node before NACT [92]. Seeds marked by Iodine-125 ($^{125}$I) can be used in the NACT setting because of a long isotope half-life (59 days). Some disadvantages are the possible limitation of this procedure by national governments, the complex organization of monitoring both patients and staff radiation exposure, and the possible depletion in the radioactive load in the case of the prolongation of NACT [31,93]. A routine pre-operative localization of radioactive seeds could be performed under mammographic or echographic guidance and, in the case of failure, via MRI. However, MRI-guided localization has some limitations, such as the presence of steel in the preloaded seed, which makes the procedure dangerous. Furthermore, in the case of using an RI technique for SLN mapping $^{125}$I, a potential interference in the signals detected by GP can occur. Although the risk is low, rupture or transection of the seeds requires an emergency iodine treatment to saturate and safeguard the thyroid gland [31,52,94].

Radar reflectors, magnetic seeds, and radiofrequency markers could alternatively be used in probe-guided techniques in which TLNB is tagged with these markers, which can be detected using a specific probe during surgery, without the need for a second preoperative procedure. Current data about the use of these techniques before NACT are limited [95]. Possible disadvantages of radar reflector-based localization are MRI artifacts, a relatively high cost, and interference with instruments in the surgery room, such as halogen lights [96,97]. Regarding magnetic and radiofrequency markers, critical aspects are concerns about their use in patients with cardiac devices (e.g., pacemakers and implantable defibrillators), significant MRI artifacts, and high costs [98–101].

In any case, most of the studies and experiences concluded that TAD, proposed to lower the FNR of SLNB alone, is not prognostically significant [102–104].

## 7. Conclusions

A large number of studies have evaluated the oncological outcomes of lymphatic mapping with SLNB in BC cN+ patients who become cN0 following NACT. In light of their results, we can state that this is a safe procedure in such patients, showing a good prognosis and low axillary failure rates. Moreover, TAD, which is the most important technique. proposed to minimize FNR, has not demonstrated a prognostic advantage over SLNB alone.

As a result, axillary surgery is moving towards a de-escalation, and some ongoing studies are attempting to assess whether a reduction in axillary adjuvant RT is possible.

Many techniques have been approved for SLNB mapping, and each one demonstrates a good detection rate in patients treated with NACT, with no significant differences between each other.

## 8. Future Directions

In the future, the identification of patients who do not require overtreatment may be possible thanks to the progress which has been made in personalized therapy. In selected BC cases, the optimal tumor response following NACT may lower or even remove the need for surgery, and subsequent therapy may have to consider the tumor biological

characteristics and not nodal status. However, we await the completion of additional prospective clinical studies evaluating different approaches.

**Author Contributions:** Conceptualization, G.F. and S.S.; methodology, G.F., A.N., S.S. and E.F.; writing—original draft preparation, G.F., A.N. and S.S.; writing—review and editing, E.F., A.R., A.M. (Alessandro Mignone), F.G., A.M. (Augusto Manzara) and G.P.; project administration, G.F. and S.S. All authors have read and agreed to the published version of the manuscript.

**Funding:** This research received no external funding.

**Data Availability Statement:** Trial data cited can be consulted on the following sites: https://www.eubreast.com/?Trials/AXSANA (accessed on 1 June 2023); https://www.clinicaltrials.gov/study/NCT01872975 (accessed on 1 June 2023); https://www.clinicaltrials.gov/study/NCT01901094 (accessed on 1 June 2023).

**Conflicts of Interest:** The authors declare no conflict of interest.

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
