# Peer review of "The Role of Sentinel Lymph Node Biopsy in Breast Cancer Patients Who Become Clinically Node-Negative Following Neo-Adjuvant Chemotherapy: A Literature Review"

_curroncol, doi:10.3390/curroncol30100630_

Round 1

Reviewer 1 Report (New Reviewer)

The authors present an interesting review on the actual topic of breast cancer surgery - the axillary staging after NAC. I recommend accepting the article after minor revisions.

1. Abstract - The letter b in "based" should be capitalised.

2. Introduction -  "Advantages of this technique are the minimally invasive approach and the possibility of breast preservation."  How can SLNB affect breast preservation?

3. Introduction - "The aim is to favor breast-conservative surgery over ALND." Again, you are mixing apples and oranges - almost every breast cancer surgery has two phases - breast surgery and axillary surgery, here you are comparing breast-conservation and ALND - possibly instead of SLNB and ALND. 

4. Introduction -  "Working Group for Gynecological Oncology (AGO Breast Committee) accepts TAD and ALND as recommended strategies [28]." So which one is recommended - TAD or ALND? Or you should specify in which situation they recommend ALND.

5. Introduction - lines 80-81 - TAD could be marked by various markers, not just tattoo or clips. Here are some tips for citation about markers in TAD:

https://pubmed.ncbi.nlm.nih.gov/36831516/

https://pubmed.ncbi.nlm.nih.gov/37596658/

6. Results - line 130 - typing error in the word "three".

7. Discussion - lines 308-309 - Again, I recommend to review the techniques for lymph node marking in TAD.

8. Discussion - The authors should explain the difference between TAD and TLNB making it more clear for readers.

9. Discussion - line 360-361 - The main goal of TAD and TLNB is lower FNR compared to SLNB, this information should be mentioned instead of the last sentence. Also, this information is missing in the Conclusion.

Author Response

Thank you for reviewing our review entitled "The role of sentinel lymph node biopsy in breast cancer patients who become clinically node-negative following neo-adjuvant chemotherapy: A Literature Review”. Thank you for the comments and suggestions. We have responded to each comment below.

  1. Abstract - The letter b in "based" should be capitalised.

Response 1: we have capitalised letter b.

  1. Introduction - "Advantages of this technique are the minimally invasive approach and the possibility of breast preservation."  How can SLNB affect breast preservation?

Response 2: we have revised the sentence.

  1. Introduction - "The aim is to favor breast-conservative surgery over ALND." Again, you are mixing apples and oranges - almost every breast cancer surgery has two phases - breast surgery and axillary surgery, here you are comparing breast-conservation and ALND - possibly instead of SLNB and ALND. 

Response 3: we have revised the sentence.

  1. Introduction - "Working Group for Gynecological Oncology (AGO Breast Committee) accepts TAD and ALND as recommended strategies [28]." So which one is recommended - TAD or ALND? Or you should specify in which situation they recommend ALND

Response 4: we have better explained which are the recommended strategies.

  1. Introduction - lines 80-81 - TAD could be marked by various markers, not just tattoo or clips. Here are some tips for citation about markers in TAD:

https://pubmed.ncbi.nlm.nih.gov/36831516/

https://pubmed.ncbi.nlm.nih.gov/37596658/

Response 5: we have deepened the topic and we have also included the suggested references.

  1. Results - line 130 - typing error in the word "three".

Response 6: we have corrected the word.

  1. Discussion - lines 308-309 - Again, I recommend to review the techniques for lymph node marking in TAD.

Response 7: we have deepened the topic.

  1. Discussion - The authors should explain the difference between TAD and TLNB making it more clear for readers.

Response 8: we have integrated text with a full explanation of differences between TAD and TLNB.

  1. Discussion - line 360-361 - The main goal of TAD and TLNB is lower FNR compared to SLNB, this information should be mentioned instead of the last sentence. Also, this information is missing in the Conclusion.

Response 9: we have integrated the suggested information both in discussion and in conclusion.

Reviewer 2 Report (New Reviewer)

Dear Authors,

Overall good, but some minor comments : 

1) All abreviations - please spell out.

2) there is a lack of diagram presenting the results. 

3) This huge group of patients in this review - should results be presented with statystical tests and as a meta-analysis or systemic review not just review ? It would be more readeable for me and with PRISMa diagram (flow chart) - please consider to change the article to the Systemic Review with PRISMA flow chart. 

4) Please consider this reference :DOI: 10.3390/curroncol29050235

5) Scientific soundness - OK. 

All in all, some minors comments - authors probably will well address all the suggestions.

Sincerely yours.

Author Response

Thank you for reviewing our review entitled "The role of sentinel lymph node biopsy in breast cancer patients who become clinically node-negative following neo-adjuvant chemotherapy: A Literature Review”. Thank you for the comments and suggestions. We have responded to each comment below

1) All abreviations - please spell out.

Response 1: we have spelt out abbreviations.

2) there is a lack of diagram presenting the results. 

Response 2: In Table 1, Table 2 and Table 3 the most important characteristics and results, extracted from all included studies in the present review, are presented.

3) This huge group of patients in this review - should results be presented with statystical tests and as a meta-analysis or systemic review not just review? It would be more readeable for me and with PRISMa diagram (flow chart) - please consider to change the article to the Systemic Review with PRISMA flow chart. 

Response 3: we have updated the search, we have systematically reviewed articles and we have inserted a flow chart, as you suggested.

4) Please consider this reference: DOI: 10.3390/curroncol29050235

Response 4: we inserted suggested reference.

5) Scientific soundness - OK. 

Response 5: Thanks

Reviewer 3 Report (New Reviewer)

Because of regional difference in recommendation of lymph node surgery, the oncological consequence of sentinel lymph node biopsy (SLNB) in breast cancer patients treated with neoadjuvant chemotherapy (NACT) is controversial. In this review, authors summarized latest literature (2018 to Jun 2023) to evaluate the impact of SLNB in patients that are initially cN+ and become cN0 after NACT. The safety and effectiveness of SLNB has been demonstrated by numerous trials. To provide additional value to scientific community, in-depth discussion is required. Please see following comments.

Even though the literature search is thorough and outline of evidence is well-orchestrated, discussion of biological impacts of axillary lymph node dissection (ALND) and irradiation therapy (RT), such as tissue damage, altered drainage and inflammatory response is lacking. Moreover, the downstaging frequency of various subtypes of breast cancer and corresponding NACT approach shall be listed, and the potential reason can be further discussed. At last, in addition to comparing primary outcome like recurrence free survival across trials, notable alteration in complication frequency (SLNB or RT vs ALND) can be supplemented to the text.

Author Response

Thank you for reviewing our review entitled "The role of sentinel lymph node biopsy in breast cancer patients who become clinically node-negative following neo-adjuvant chemotherapy: A Literature Review”. Thank you for the comments and suggestions. We have responded to each comment below. 

1) Even though the literature search is thorough and outline of evidence is well-orchestrated, discussion of biological impacts of axillary lymph node dissection (ALND) and irradiation therapy (RT), such as tissue damage, altered drainage and inflammatory response is lacking.

Response 1: we have deepened the topic as you suggested.

2) Moreover, the downstaging frequency of various subtypes of breast cancer and corresponding NACT approach shall be listed, and the potential reason can be further discussed.

Response 2: we have deepened the topic as you suggested.

3) At last, in addition to comparing primary outcome like recurrence free survival across trials, notable alteration in complication frequency (SLNB or RT vs ALND) can be supplemented to the text.

Response 3: we have found alteration in complication frequency (SLNB vs ALND) in some of trials included in the systematic review. Information and data have been supplemented to the text, as you suggested.

We have not found data about the different RT complication frequency across trials systematically reviewed.

Round 2

Reviewer 1 Report (New Reviewer)

The Authors repaired their manuscript according to the suggestions. I recommend to publish the article. 

One more suggestion - consider to discuss the complications after TAD in section F (Effect on complications). 

Reviewer 2 Report (New Reviewer)

Accept as it is

Reviewer 3 Report (New Reviewer)

Authors added another paragraph to address the related complication and enhanced complication rate after axillary lymph node dissection. I understand this is a clinical review yet the fundamental biology behind the clinical manifestations should be considered. Nevertheless, this manuscript meets the standard for publication.

This manscript is a resubmission of an earlier submission. The following is a list of the peer review reports and author responses from that submission.

Round 1

Reviewer 1 Report

This article is a rather confusing literature review where the authors propose to understand the role of SNB in cN1 BC that undergo downstaging with NACT to cN0. Although the topic is relevant, there are a series of core conceptual errors in the way the review is done and results presented, including confusing presentation of the results and some out of the scope of the review, with lengthy discussion that don't seem to lead the reader to a sensible conclusion. English sometimes is confusing too.

Author Response

Thank you for reviewing our narrative review entitled "The role of sentinel lymph node biopsy in breast cancer patients who become clinically node-negative following neo-adjuvant chemotherapy: A Literature Review”. Thank you for the comments and suggestions. We have responded to comment below.

Response

We reorganized our narrative review, in an attempt to improve the quality of the review. English quality was revised.

Reviewer 2 Report

The authors of this manuscript have applied a literature review study on the role of sentinel lymph node biopsy in breast cancer patients who become clinically node-negative following neo-adjuvant chemotherapy. This title is worthy of investigation. However, regarding reviewing the manuscript, some comments are mentioned below that need more clarification, modification, or explanation.

Abstract:

1.       It is better that abstract be structured into background, objective, methods, results, and conclusion.

Introduction:

1.       It’s better to define “sentinel lymph node” in a sentence first and compare it with regional lymph nodes.

2.       In lines 64 and 87, the references that are provided in the text, should be transferred to the “References “section.

Methods:

1.       In 3nd row of table 1, SLBN should be changed into SLNB.

2.       Table 2 is disorganized from its middle. Please revise it.

3.       Mention the methods used to assess risk of bias in the included studies.

4.       Describe any methods used to assess certainty (or confidence) in the body of evidence for an outcome.

5.       Mention any statistical method or statistical analysis that was applied for the study.

Part 3:

1.       In this part, every study was discussed in one paragraph. This type of writing is not acceptable and should be revised. It is better to consider a variable (e.g. DFS, OS) in a paragraph and afterwards explain and discuss numerous studies about that variable.

Part 4:

1.       AMAROS in line 227 should be mentioned in its complete form as well.

2.       Websites that are mentioned in lines 239 and 240 should be transferred to “References” section.

Part 5:

1.       Again in this part, every study was discussed in one paragraph. It’s better to discuss every SLN mapping method (e.g. RI, BD, ICG, etc.) in a single paragraph and discuss different aspects, advantages, and disadvantages of that particular method in that paragraph.

2.       Throughout all the manuscript, the reviewed studies are usually presented in such a detail that is not actually necessary. In other words, it is not attracting to report all the numbers and results that a study has achieved. It is better that reporting of the studies be concise and selective without mentioning the trivial things. Only the most important and straightforward part of a study in needed to be reported.

Other comments:

1.       Describe the results of the search and selection process, from the number of records identified in the search to the number of studies included in the review, ideally using a flow diagram.

2.       Cite studies that might appear to meet the inclusion criteria, but which were excluded, and explain why they were excluded.

3.       Discuss any limitations of the evidence included in the review.

4.       Discuss any limitations of the review process used.

5.       Discuss implications of the results for practice, policy, and future research.

Wish you luck and prosperity in preparing the manuscript.

Although the quality of English language is acceptable generally, minor editing of English language required.

Reviewer 3 Report

The paper from Ferrarazzo and coworkers reports available evidence on feasibility and reliability of SN biopsy in early breast cancer patients with cN+ disease treated with neoadjuvant chemotherapy. The paper is a complete review of available data from published trials, reassures practicing physicians on the safety of this procedure which is not uniformly endorsed by international guidelines. The authors in the discussion might acknowledge some limitations of available evidence: 1) trials have included mixed patient population (HR+, HER2+, TN), 2) new effective postoperative treatments are available for patients not achieving a pCR with NACT and this might require a more stringent definition of acceptable FNR.

Author Response

Thank you for reviewing our narrative review entitled "The role of sentinel lymph node biopsy in breast cancer patients who become clinically node-negative following neo-adjuvant chemotherapy: A Literature Review”. Thank you for the comments and suggestions. We have responded to comment below.

Response

We have added some limitations of available evidence and of our review.

Round 2

Reviewer 2 Report

Dear authors of the manuscript,

Your effort for reviewing the manuscript is appreciated. However, there are lots of errors in writing the manuscript that have not been revised yet. The result and discussion part of the manuscript are still lengthy and disorganized. The findings from other studies were not combined to properly achieve a great and comprehensible manuscript. On the contrary, the quotations from other studies were mentioned directly without deleting the trivial things. Besides, the narrative type of the study was not mentioned anywhere, including the title of the manuscript. Moreover, a practical chart or table is not available to get the concise findings of the manuscript in a glance. It seems completely necessary in writing your manuscript. 

with best wishes

Minor revision of English language utilized in the manuscript is needed.

Author Response

Response to Reviewer 2 Comments.

Reviewer 2 Comments: Your effort for reviewing the manuscript is appreciated. However, there are lots of errors in writing the manuscript that have not been revised yet. The result and discussion part of the manuscript are still lengthy and disorganized. The findings from other studies were not combined to properly achieve a great and comprehensible manuscript. On the contrary, the quotations from other studies were mentioned directly without deleting the trivial things. Besides, the narrative type of the study was not mentioned anywhere, including the title of the manuscript. Moreover, a practical chart or table is not available to get the concise findings of the manuscript in a glance. It seems completely necessary in writing your manuscript. 

Minor revision of English language utilized in the manuscript is needed.

Response

Thank you for reviewing the manuscript, for your suggestions and corrections.

We tried to improve the quality of the text and tables. In particular:

  1. In Ln 21 and ln 95 the narrative type of the review has been mentioned;
  2. We rearranged paragraphs 3,4 and 5. We added new and better organized tables and deleted the trivial things. In the main text we inserted the important information and data to explain and discuss.
  3. English language has been checked.

King Regards.

Dr. Giulia Ferrarazzo